# Statistical estimation of probable maximum precipitation

Anne Martin[1], Elyse Fournier[2], and Jonathan Jalbert[1]

[1]Department of mathematics and industrial engineering, Polytechnique Montréal, Canada
[2]Direction of expertise, engineering and standardization - Dam safety and infrastructure, Hydro-Québec, Canada
**Correspondence:** Jonathan Jalbert (jonathan.jalbert@polymtl.ca)

**Abstract.**

Civil engineers design infrastructure exposed to hydrometeorological hazards, such as hydroelectric dams, using probable maximum precipitation (PMP) estimates. The World Meteorological Organization (WMO) defines PMP as *the maximum amount of water that can physically accumulate over a given time period and region, depending on the season and without considering long-term climate trends*. Current PMP estimation methods have several flaws: some required variables are not directly observable and rely on a series of approximations; uncertainty is not always accounted for and can be complex to quantify; climate change, which exacerbates extreme precipitation events, is difficult to incorporate; and subjective choices increase estimation variability. In this paper, we derive a statistical model from the WMO's PMP definition and use it for estimation. This novel approach leverages the Pearson Type I distribution, a generalization of the Beta distribution over an arbitrary interval, allowing for uncertainty quantification and the incorporation of climate change effects. Several estimation procedures are considered, including the method of moments, maximum likelihood, and Bayesian estimation. However, statistical PMP estimation remains challenging because a short-tailed model is applied to typically heavy-tailed precipitation data. The performance of the proposed approach is assessed through a simulation study and applied to actual precipitation data from two nearby stations in Canada. Finally, we provide and discuss recommendations for best practices in PMP estimation.

## 1 Introduction

### 1.1 Context

Dams play a crucial role in regulating streamflow and generating hydroelectricity, providing essential water management and renewable energy resources. Over-sizing these infrastructures during construction or renovation can lead to unnecessary costs. Conversely, under-sizing them can pose risks to dam safety, the environment, and surrounding populations, and may result in excessive costs. In the province of Québec (Canada) alone, there are over 6,000 dams over one meter in height spread across the region (CEHQ, 2023). According to regulations derived from the Dam Safety Act, civil engineers must design these structures using an estimation of the local extreme flood conditions. Depending on the risk in case of breakage, various flood estimations are used, such as the millennial, decamillennial, or probable maximum flood (PMF). The latter is the greatest theoretically possible flood on a specific watershed and is computed based, among other factors, on the probable maximum precipitation (PMP). According to the World Meteorological Organization (WMO, 2009), the PMP corresponds

to the maximum precipitation accumulation over a fixed duration in a given region. Several PMP estimation techniques have been developed, including moisture maximization, the empirical Hershfield approach, and approaches based on extreme value theory. In general, PMP estimation is challenging and sensitive to the data. On the one hand, uncertainty and climate change effects are difficult to incorporate into non-statistical methods, and commonly used moisture maximization approaches involve several subjective judgments. On the other hand, statistical PMP estimation is challenging because its definition assumes a bounded tail, whereas precipitation data suggests an unbounded tail (see, e.g., Martins and Stedinger, 2000). The following sections summarize the different approaches to PMP estimation.

## 1.2 Estimation based on moisture maximization

In its 2009 manual, the WMO details several PMP estimation approaches, with hydrometeorological methods combining algorithms of storm selection, transposition, and maximization being the most popular in Canada. In regions where snow cover is important enough for the PMF to result from a combination of the PMP and snowmelt, the WMO typically recommends estimating both spring and summer-fall PMPs. Regarding storm selection, some authors use all rain events where the precipitation height exceeds a given threshold (Beauchamp et al., 2013), while others utilize all observed precipitation data over a given period (Ben Alaya et al., 2018). This selection process is usually carried out by meteorologists and depends on physical factors (CEHQ and SNC-Lavalin, 2004; DTN and MGS Engineering, 2020; Environmental Water Resources Group Ltd., 2020). Since the number of selected storms is small and varies from one calculation to another and among different meteorologists, this selection process introduces significant variability in the estimation of the PMP.

To increase the number of storms used in PMP estimation, a common practice is to include storms from neighboring areas that are likely to also affect the region of interest. Over the past decades, meteorologists have developed various techniques considering the orography and other features of the areas to realistically transpose storms (WMO, 2009). This storm transposition can be incorporated into the storm selection process of PMP estimation methods. While it increases the sample size for PMP estimation, it also introduces additional sources of variability and subjectivity.

The moisture maximization approach estimates the PMP using the relationship between the amount of precipitation and the humidity of the air. Let $Y_i$ be the precipitation of storm $1 \leq i \leq n$ among the $n$ selected storms. The PMP estimation is based on moisture maximization (WMO, 2009) as follows:

$$PMP = \max_{i \in \{1, \dots, n\}} \left\{ Y_i \times \frac{PW_{\max}}{PW_i} \right\};$$  (1)

where $PW_i$ corresponds to the precipitable water of storm $i$, and $PW_{\max}$ to the maximum precipitable water. The quantity $Y_i \times \frac{PW_{\max}}{PW_i}$ is often referred to as the maximized precipitation of event $i$ if the maximal precipitable water were available at the moment of the storm. The PMP then corresponds to the maximum of the maximized precipitation. The ratio $\frac{PW_{\max}}{PW_i}$ is referred to as the maximization ratio and is sometimes arbitrarily set to a numerical value between 1.5 and 2.5 to avoid the overestimation of the PMP (Schreiner and Riedel, 1978; Hansen, 1988; WMO, 2009; Beauchamp et al., 2013). The use of this threshold is subjective and lacks physical or mathematical justification (Rouhani and Leconte, 2016).

The moisture maximization expressed in Eq. (1) can also be rewritten as follows, given a slightly different interpretation:

$$PMP = \max_{i \in \{1,...,n\}} \left\{ \frac{Y_i}{PW_i} \times PW_{\max} \right\}. \tag{2}$$

In this last expression, the ratio $\frac{Y_i}{PW_i}$ corresponds to the ratio of precipitation to precipitable water and is referred to as the precipitation efficiency of storm $i$. The PMP occurs when the maximum precipitation efficiency coincides with the maximum precipitable water. Ben Alaya et al. (2018) utilize this definition to model the dependence between extreme values of precipitation efficiency and precipitable water. They demonstrate that the comonotonicity imposed by Eq. 2 leads to overestimation of the PMP in North America.

In practice, the precipitable water $PW_i$ at the moment of storm $i$ and the maximum amount of precipitation for the considered region, $PW_{\max}$, are unknown and must be estimated for using the moisture maximization approach. The amount of precipitable water can be estimated using the specific humidity of the air column above the area (WMO, 2009). However, for the majority of meteorological stations, specific humidity is neither observed nor recorded. The recommended estimation of precipitable water by the WMO (2009) uses the dew point, which is usually recorded, and requires pseudo-adiabatic conditions (United 70 States Weather Bureau, 1960; Miller, 1963). Viswanadham (1981) observed that the relation between surface dew point and precipitable water is greater when the latitude is over 25° than in lower latitude zones. A study conducted by Chen and Bradley (2006) in the Chicago region indicates that the pseudo-adiabatic conditions hypothesis could lead to overestimation of precipitable water, and Rouhani and Leconte (2020) noted that PMP estimates vary greatly depending on how the precipitable water was approximated. The uncertainty of these estimations is often neglected in PMP estimation. It is not uncommon for 75 the uncertainty of $PW_i$ to lead to a precipitation efficiency larger than 1, which is physically impossible.

## 1.3 Estimation based on Hershfield's empirical approach

The Hershfield empirical method (Hershfield, 1961a, b, 1965) is an alternative to moisture maximization for PMP estimation. The method relies on a series of $m$ precipitation annual maxima. The PMP estimates is as follows:

$$PMP = \bar{x} + Ks \tag{3}$$

where $\bar{x}$ and $s$ correspond respectively to the mean and the standard deviation of the series of annual maxima and $K$ corresponds to the frequency factor for estimating PMP at that location. Hershfield (1961a) proposed a method to estimate this factor. This approach is widely employed for its simplicity and ease of use but can only estimate PMP over smaller watersheds (WMO, 2009). It also has the advantage of not requiring additional hydrometeorological data such as specific humidity or dew point. Only precipitation series from which annual maxima are extracted are required.

This method is often classified as a statistical technique, but in this paper, it is considered empirical due to the nature of the link between the PMP and the sample moments. It should be noted that the PMP durations considered by Hershfield (1965) and available in WMO (2009) are all less than or equal to 24 hours, which is inadequate for calculating longer-duration PMP.

## 1.4 Estimation based on Extreme value theory

The relevance of the Hershfield procedure can also be questioned. Eq. 3 defines the PMP as an extreme quantile of the distribution, estimated using only the mean and standard deviation. This approach relies on the bulk of the distribution to extrapolate into the tail, which is inherently hazardous. Extreme value theory (EVT, see e.g. Coles, 2001) is a branch of statistics that focuses on extreme values. It provides asymptotic parametric distributions (the Generalized Extreme value and the Generalized Pareto distributions) and rigorous frameworks (block maxima and peaks-over-threshold) for extrapolating beyond the range of observations.

As a statistical approach, it is easier to incorporate non-stationarity induced by climate change and to provide uncertainty in the estimates. However, extreme value analysis suggests that the precipitation distribution is unbounded, which is inconsistent with the PMP definition. To reconcile the PMP definition with extreme value theory, some authors propose using a very large return period as the PMP estimate. For example, Koutsoyiannis (1999) shows that PMP estimates obtained through the Hershfield method correspond, on average, to return periods of 60,000 years when estimated using EVT. National Academies of Sciences, Engineering, and Medicine (2024) also suggest using a quantile of an extreme-value distribution corresponding to an *extremely low annual probability of being exceeded.*

## 1.5 Estimation using simulated data

The methods of estimation of the PMP presented by the WMO (2009) only consider precipitation data observed at hydrometeorological stations. Methodologies adapted for regional simulations have been developed for regions in Canada (Beauchamp et al., 2013; Rousseau et al., 2014; Clavet-Gaumont et al., 2017; Rouhani and Leconte, 2018, 2020), North America (Kunkel et al., 2013; Ben Alaya et al., 2018, 2020a, b) and other parts of the world (Sarkar and Maity, 2020; Visser et al., 2022).

The use of these climate simulations not only allows for the consideration of a greater number of extreme rainfall events but also enables the estimation of futures PMP. Indeed, the WMO (2009) defines the PMP as stationary values and CC isn't taken into account in the calculations. However, it is widely acknowledged that CC has a direct impact on extreme precipitation events and should therefore be considered in their estimation. Using projected climate simulations, Kunkel et al. (2013) demonstrate a global increase of water vapor concentration in the atmosphere, without a sufficient evolution in values of upward vertical motion or horizontal wind speed, factors that could counterbalance the rise in air humidity. This increase implies larger future values of $PW_{max}$, and consequently, an increase in PMP. Several papers conclude that PMP will generally increase in future climate (Beauchamp et al., 2013; Rousseau et al., 2014; Clavet-Gaumont et al., 2017; Rouhani and Leconte, 2018, 2020; Ben Alaya et al., 2018, 2020a, b; Sarkar and Maity, 2020; Visser et al., 2022).

## 1.6 Objectives of the paper

PMP estimations using the non-statistical approaches described in the previous sections are highly sensitive to arbitrary choices and are generally provided without accounting for uncertainty. Statistical approaches, on the other hand, facilitate the inclusion of non-stationarity and the quantification of uncertainty. However, reconciling the PMP definition as the upper bound of

|  | Montréal | St-Hubert |
| --- | --- | --- |
| Period | 1953–2024 | 1949–2024 |
| Number of days with precipitation | 5321 | 5303 |
| Mean of non-zero precipitation | 6.9 mm | 7.4 mm |
| Maximum precipitation | 81.9 mm | 106.5 mm |
| Mean of precipitation annual maxima | 44.9 mm | 49.9 mm |
| Standard deviation of precipitation annual maxima | 14.3 mm | 18.0 mm |
| Series autocorrelation (lag of one day) | 0.0092 | 0.0095 |

**Table 1.** Summer (May to October) daily precipitation statistics for the Montréal and St-Hubert stations.

an unbounded distribution remains particularly challenging. The objective of this paper is to develop a statistical model for PMP estimation based on the WMO definition, which assumes an upper bound. As a statistical approach, it offers two key advantages: (1) uncertainty is quantifiable, and (2) most subjective choices are eliminated, enhancing the reproducibility of the estimation.

The remainder of the paper is organized as follows: Section 2 describes the data used for the proposed method and the current PMP estimations at the selected locations. Section 3 presents the proposed statistical model for estimating the PMP, and Section 4 provides a simulation study to assess the model's performance in PMP estimation. The PMP estimations for real datasets are presented and discussed in Section 5. Our recommendation on PMP estimation is provided in Section 6, and Section 7 concludes the paper. The data and code for reproducing all figures and results are available at the following public repository: https://github.com/JuliaExtremes/PMP.jl.

## 2 Data

### 2.1 Observations

The proposed method for estimating the PMP is demonstrated using data from two meteorological stations in Québec, Canada, located 26 km apart: the Montréal Pierre-Elliott-Trudeau International Airport station (1953–2024) and the St-Hubert Airport station (1949–2024). The data are available from the Environment and Climate Change Canada (ECCC) website. Daily precipitation (in mm) and dew point (in Celsius) were extracted from May 1 to October 31 each year to focus on liquid rainfall and minimize the effect of seasonality. While it may still be present, it appears negligible compared to the natural variability of precipitation and precipitable water, as illustrated in Figure 1 for the Montréal data. Descriptive statistics of recorded precipitation at these two stations are provided in Table 1. Figure 2 shows the histogram of non-zero daily rainfall for each station. Typically, for precipitation at the considered locations, autocorrelation exists in daily non-zero series but is very weak (0.0092 for Montréal and 0.0095 for St-Hubert) and short-range.

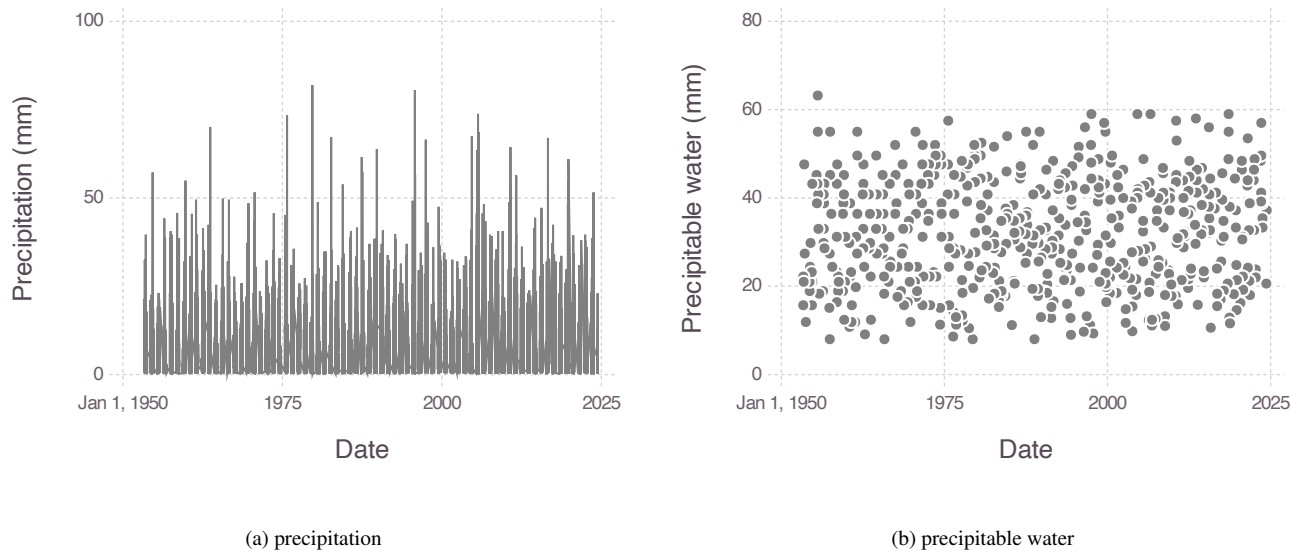

(a) precipitation

(b) precipitable water

**Figure 1.** Time series of (a) daily precipitation and (b) precipitable water for the top 10% of storms recorded at the Montréal station.

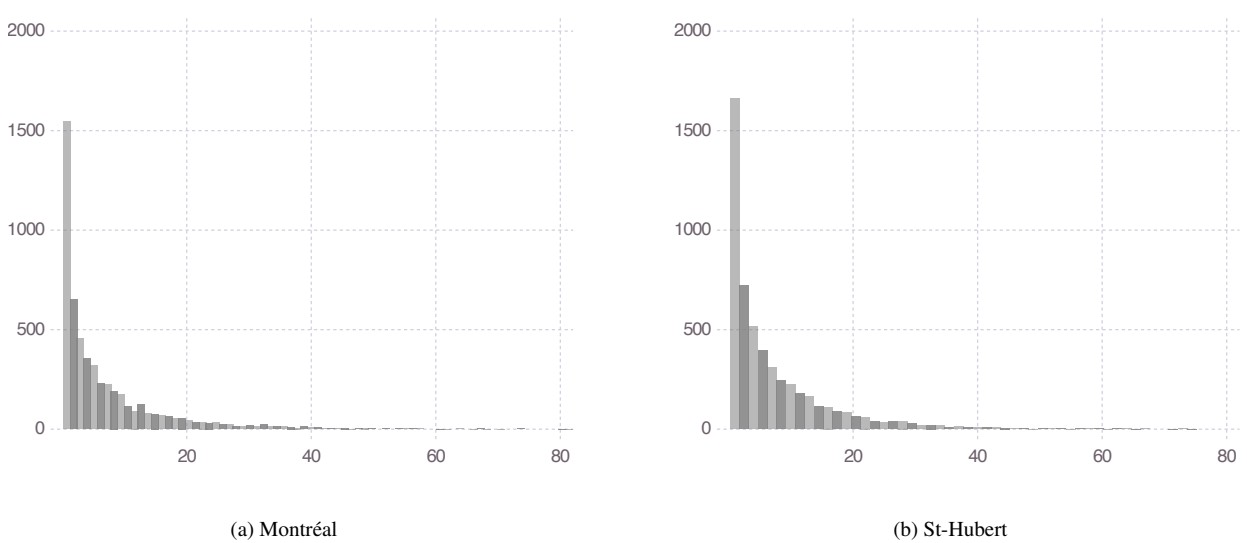

(a) Montréal

(b) St-Hubert

**Figure 2.** Histogram of the non-zero summer precipitation in mm for (a) Montréal (QC) and (b) St-Hubert (QC).

| Approach | Montréal | St-Hubert |
|---|---|---|
| Moisture maximization | 282 mm | 436 mm |
| Hershfield methods using $K = 15$ | 261 mm | 322 mm |
| Extreme value analysis | 185 mm | 200 mm |

**Table 2.** Estimated PMP at Montréal and St-Hubert using the standard approaches.

## 2.2 PMP estimates

As a point of comparison, summer–fall PMP estimates for both stations are calculated using the moisture maximization method, Hershfield's approach, and the 60,000-year return level estimated with EVT. For the moisture maximization method, daily precipitation amounts from the top 10% of storms for each year are selected, as proposed by Clavet-Gaumont et al. (2017).

A sensitivity analysis was performed on the storm selection percentage (10%, 1%, or 0.1%), but the PMP estimates remained unchanged because the maximized events were the same for both locations. Since precipitable water was not directly observed, it was estimated using the dew point over twelve hours, as described by WMO (2009), which may have affected the quality of the PMP estimates. The corresponding PMP estimates for both stations are provided in Table 2. The PMP estimates are 282 mm and 436 mm for Montréal and St-Hubert, respectively, corresponding to maximization ratios of 4.4 and 4.9. Some

authors suggest limiting this ratio to a value between 1.5 and 2.5 to constrain PMP estimation. Setting the maximization ratio to 2.0 would yield PMP estimates of 128 mm and 178 mm for Montreal and St-Hubert. While this would improve the moisture maximization results, it merely conceals the methodology's flaws, particularly its high variability, rather than addressing them.

Note that for this approach, it is also possible to estimate the PMP using the 100-year return level of precipitable water instead of the sample maxima (e.g. Ben Alaya et al., 2018), but with our data, the estimated PMP values were similar: 284 mm

and 427 mm for Montréal and St-Hubert, respectively.

For the Hershfield's approach, the frequency factor of $K = 15$ is employed as proposed by Hershfield (1961a). The adjustment based on the number of data points, as suggested by WMO (2009), is unnecessary. Hershfield's PMP estimates are 261 mm for Montréal and 322 mm for St-Hubert, and are also provided in Table 2.

The 60,000-year return level was estimated using the Peaks-Over-Threshold model (see, e.g., Coles, 2001; Davison and

160 Smith, 1990). The estimated return levels are 185 mm and 200 mm for the Montréal and St-Hubert data respectively.

## 3 Methodology

In this section, a statistical model is developed for PMP estimation based on the definition expressed in Eq. (1). Statistical inference methods for this proposed model are also described.

### 3.1 Statistical model

Starting with the principles underlying moisture maximization (WMO, 2009), we develop a sensible statistical model that assumes an upper bound for precipitation. From Eq. (1), let $\tilde{Y}_i$ denote the maximized daily precipitation on day $i$:

$$\tilde{Y}_i = \frac{Y_i}{PW_i} \times PW_{\max}. \tag{4}$$

Factoring for the actual precipitation of day $i$ gives the following expression:

$$Y_i = \frac{PW_i}{PW_{\max}} \times \tilde{Y}_i, \tag{5}$$

where $PW_{\max}$ is the maximum of precipitable water. Hence, $0 < \frac{PW_i}{PW_{\max}} \leq 1$. Since the PMP corresponds to the maximum of the maximized precipitations $\tilde{Y}_i$, the maximized precipitation can be viewed as a fraction of the PMP, $i.e.$, $\tilde{Y}_i = r_i PMP$, where $0 < r_i \leq 1$. It is then possible to express the actual daily precipitation as follows:

$$Y_i = \frac{PW_i}{PW_{\max}} \times r_i PMP. \tag{6}$$

The ratio $\frac{PW_i}{PW_{\max}} \times r_i$ lies in $(0,1]$ since each of the multiplicative factors is within $(0,1]$. This ratio can naturally be modeled using the Beta distribution, a flexible distribution for a random variable taking values in the unit interval. Actual precipitation $Y_i$, which corresponds to this ratio multiplied by the PMP, can be modeled using the Beta distribution rescaled to the interval $(0, PMP)$. The Beta distribution on the interval $(a, b)$ for $a < b$ is referred to as the Pearson Type I distribution (Johnson et al., 1995, Chap. 24).

Therefore, we propose to model the actual precipitation of day $i$ with the Pearson Type I distribution as follows:

$$Y_i \sim PearsonType1(0, \psi, \alpha, \beta); \tag{7}$$

where the Beta parameters $\alpha > 0$ and $\beta > 0$ govern the ratio $\frac{PW_i}{PW_{\max}} \times r_i$ and where $\psi > 0$ corresponds to the PMP. The lower bound is set at 0 because only non-zero precipitation events are considered. When $0 < \alpha < 1$ and $\beta \geq 1$, the Pearson Type I density is monotonically decreasing and convex, resembling the precipitation histogram shown in Figure 2. With this proposed statistical model, the PMP constitutes a distribution parameter to be estimated with the data. Uncertainty can then be provided using the usual statistical methods, as described in the next section.

### 3.2 Parameter estimation

Three methods are considered for estimating the parameters of the proposed model: the method of moments, maximum likelihood estimation, and the Bayesian method.

### 3.2.1 Method of moments

The first four central moments, namely the mean $m$, the variance $v$, the skewness $s$, and the kurtosis $k$, of the Pearson Type I distribution $PearsonTypeI(0, \psi, \alpha, \beta)$ are given by the following expressions (adapted from Johnson et al., 1995, Chap. 24):

$$m = \frac{\alpha\psi}{\alpha+\beta}; \tag{8}$$

$$v = \frac{\alpha\beta\psi^2}{(\alpha+\beta)^2(\alpha+\beta+1)}; \tag{9}$$

$$s = \frac{2(\beta-\alpha)\sqrt{\alpha+\beta+1}}{(\alpha+\beta+2)\sqrt{\alpha\beta}}; \tag{10}$$

$$k = \frac{6(\alpha^3 - \alpha^2(2\beta-1) + \beta^2(\beta+1) - 2\alpha\beta(\beta+2))}{\alpha\beta(\alpha+\beta+2)(\alpha+\beta+3)} + 3. \tag{11}$$

The skewness and kurtosis depend only on the shape parameters $\alpha$ and $\beta$, and not on the upper bound $\psi$. It is possible to invert these equations to factorize for the distribution parameters (adapted from Johnson et al., 1995, Chap. 24):

$$\alpha = -\frac{s(k+3) + \left(\frac{-s(k+3)-\sqrt{s^2(k+3)^2-4(2k-3s^2-6)(4k-3s^2)}}{2(2k-3s^2-6)}\right)(10k-12s^2-18)}{\sqrt{(s(k+3))^2-4(2k-3s^2-6)(4k-3s^2)}} + 1; \tag{12}$$

$$\beta = -\frac{s(k+3) - \left(\frac{-s(k+3)+\sqrt{s^2(k+3)^2-4(2k-3s^2-6)(4k-3s^2)}}{2(2k-3s^2-6)}\right)(10k-12s^2-18)}{\sqrt{(s(k+3))^2-4(2k-3s^2-6)(4k-3s^2)}} + 1; \tag{13}$$

$$\psi = a + \sqrt{v}\left(\frac{\sqrt{s^2(k+3)^2-4(2k-3s^2-6)(4k-3s^2)}}{(2k-3s^2-6)}\right). \tag{14}$$

To estimate the parameters of the Pearson Type I distribution using the method of moments from a random sample, the empirical moments of the sample–sample mean, sample variance, sample skewness, and sample kurtosis– are plugged into Eqs. (12)–(14) to obtain the parameter estimates. The uncertainty of the parameter estimates can be assessed through non-parametric bootstrap (Efron, 1979).

### 3.2.2 Maximum likelihood

The density of precipitation $Y_i$ distributed as the $PearsonType1(0, \psi, \alpha, \beta)$ is given as follows (Johnson et al., 1995):

$$f_{(Y_i|\psi,\alpha,\beta)}(y_i) = \frac{\Gamma(\alpha+\beta)}{\Gamma(\alpha)\Gamma(\beta)} \frac{(y_i)^{\alpha-1}(\psi-y_i)^{\beta-1}}{\psi^{\alpha+\beta-1}}, \quad 0 < y_i < \psi. \tag{15}$$

Assuming that the $n$ non-zero daily summer precipitation are independent, the likelihood can be written as follows:

$$f_{(\boldsymbol{Y}|\psi,\alpha,\beta)}(\boldsymbol{y}) = \prod_{i=1}^{n} f_{(Y_i|\psi,\alpha,\beta)}(y_i), \tag{16}$$

where $\boldsymbol{Y} = (Y_i, \ldots, Y_n)$ denotes the vector of the $n$ non-zero precipitations.

Maximizing the likelihood expressed in Eq. (16) is a non-regular problem (Wang, 2005). When $\beta > 1$, a local maximum exists, allowing parameter estimates to be obtained. Additionally, parameter uncertainty can be estimated using the Fisher information matrix. However, when $\beta \leq 1$, no local maximum exists, causing the estimation procedure to fail. Several solutions have been proposed for this issue, but they are not relevant to the present paper since, for precipitation, the parameter $\beta$ is expected to be greater than 1, as precipitation density is monotonically decreasing.

The Pearson Type I distribution is continuous, as expressed in Eq. (15). However, precipitation measurements are discrete. For our data, the precipitation measurement resolution is 0.1 mm, and no precipitation less than 0.2 mm can be measured. This discretization of precipitation measurements has a larger impact on small amounts. Discrepancies appear between the continuous distribution and the discrete measurements, which places mass on points of measurement. One approach to tackle this problem if needed is to censor the likelihood function for small precipitation amounts below a given threshold $u > 0$ (e.g., Naveau et al., 2016) as follows:

$$f^c_{(\boldsymbol{Y}|\psi,\alpha,\beta)}(\boldsymbol{y}) = \prod_{\{i:y_i \leq u\}} I_{\frac{u}{\psi}}(\alpha,\beta) \prod_{\{i:y_i > u\}} \frac{\Gamma(\alpha+\beta)}{\Gamma(\alpha)\Gamma(\beta)} \frac{(y_i)^{\alpha-1}(\psi - y_i)^{\beta-1}}{\psi^{\alpha+\beta-1}}, \tag{17}$$

where $I_y(\alpha,\beta)$ denotes the regularized incomplete beta function of parameter $(\alpha,\beta)$ evaluated at $y$. Precipitation smaller than $u$ still counts in the likelihood, but their actual values are not considered. Parameter estimates can be obtained by using this censored likelihood.

Another approach would be to set the lower bound of the Pearson Type I distribution to $(0.2 - \epsilon)$ where $\epsilon > 0$. This would maintain some mass at the measurement points but could sufficiently de-emphasize the issue, allowing the continuous likelihood to serve as a good approximation of the discrete measurements. One of these methods could be used if parameter estimation by maximum likelihood is affected by the discretization of precipitation measurements.

However, in our framework, approximating discrete precipitation measurements with a continuous model does not affect the fit. Therefore, we did not use either of the two methods.

### 3.2.3   Bayesian method

Estimation of the Pearson Type I distribution can also be performed under the Bayesian paradigm. The benefit of using the Bayesian method lies in its ability to describe uncertainty. Unlike the non-parametric bootstrap and the asymptotic Gaussian convergence of maximum likelihood estimates, Bayesian inference directly provides parameter uncertainty based on the data at hand, without relying on asymptotic arguments.

Bayesian methods require a prior distribution for the model parameters. For the Pearson Type I distribution expressed in Eqs. (15) and (16), an improper non-informative prior distribution for the upper bound $\psi > 0$ and the shape parameters $\alpha > 0$ and $\beta > 0$ can be defined as follows:

$$f_{(\psi,\alpha,\beta)}(\psi,\alpha,\beta) \propto \frac{1}{\psi}\frac{1}{\alpha}\frac{1}{\beta} \quad \text{for } \psi > 0, \ \alpha > 0 \text{ and } \beta > 0. \tag{18}$$

The same prior distribution can be used with the censored likelihood expressed in Eq. (17) or with a positive lower bound.

If prior information on the upper bound (the PMP) is available, an improper semi-informative prior can be used as follows:

$$f_{(\psi,\alpha,\beta)}(\psi,\alpha,\beta) \propto f_\psi(\psi)\frac{1}{\alpha}\frac{1}{\beta} \quad \text{for } \alpha > 0 \text{ and } \beta > 0, \tag{19}$$

where the prior information on $\psi$ is modelled with the proper density $f_\psi$. If $\beta \leq 1$, the problem is non-regular and the informative prior proposed by Hall and Wang (2005) can be used to solve this issue.

The posterior distribution of the parameters is not available in analytical form for either of the proposed prior distributions. A sample from the posterior distribution can be obtained, for example, using a Gibbs sampling scheme, and inference can be performed using the generated sample.

### 3.3 Identifiability issues

When $0 < \alpha < 1$ and $\beta \geq 1$, i.e., when the density is convex, non-identifiability issue occurs between $\beta$ and $\psi$. Indeed, these parameters can compensate for each other. For example, let $Z_1 \sim \text{PearsonType1}\left(10, \frac{1}{10}, \frac{99}{10}\right)$ and $Z_2 \sim \text{PearsonType1}\left(100, \frac{1}{10}, \frac{999}{10}\right)$ be two random variables with very different upper bounds–10 for $Z_1$ and 100 for $Z_2$–but whose moments are similar, as shown in Table 3. Both variables have the same mean and approximately the same variance. Although there are slight differences in skewness and kurtosis, these differences are not large enough to overcome the sampling uncertainty of these higher-order moments estimates.

| Variable | mean | variance | skewness | kurtosis |
|:--------:|:----:|:--------:|:--------:|:--------:|
| $Z_1$ | 0.1 | 0.09 | 5.44 | 40.58 |
| $Z_2$ | 0.1 | 0.10 | 6.22 | 57.45 |

**Table 3.** Moments for the variables $Z_1$ and $Z_2$.

This non-identifiability issue is even more critical for parameter estimation using the model likelihood (both maximum likelihood and Bayesian methods). For example, consider the variable $Z_1$ again and generate a large random sample of size 5000. The log-likelihood of the model evaluated at the true parameter vector $\left(10, \frac{1}{10}, \frac{99}{10}\right)$ is 35678.3. The log-likelihood evaluated at another parameter vector $\left(100, \frac{1}{10}, \frac{999}{10}\right)$ is 35678.0, which is practically the same, even though the parameters are quite different. The impact of this non-identifiability issue is assessed for parameter estimation with the method of moments, maximum likelihood, and Bayesian method with a simulation study provided in the following section.

## 4 Simulation study

In this section, a simulation study is conducted to assess the performance of parameter estimation methods for two different distribution behaviors: concave and convex density. The Pearson Type I distribution with parameters $(0, 50, 2, 2)$ is used for the concave distribution, while the Pearson Type I distribution with parameters $(0, 50, \frac{1}{10}, 6)$ is used for the convex distribution.

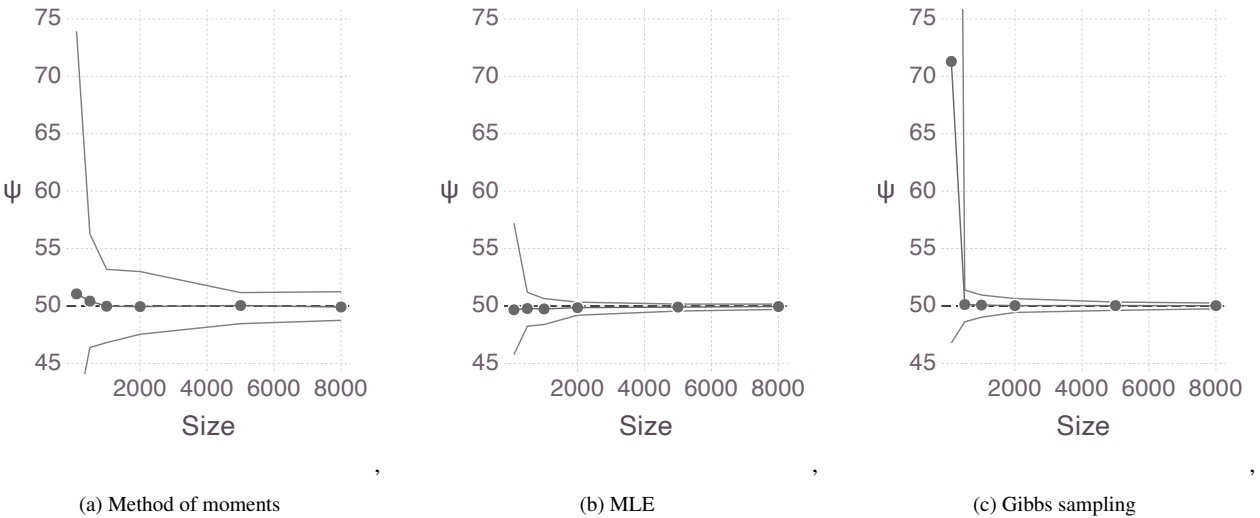

(a) Method of moments        (b) MLE        (c) Gibbs sampling

**Figure 3.** Mean and 95% empirical confidence interval for the $\psi$ estimates of the 100 samples of the $PearsonType1(0, 50, 2, 2)$ distribution obtained with (a) the method of moments, (b) the maximum likelihood and (c) the Bayesian methods using Gibbs sampling.

For each of these distributions, 100 random samples of various sizes were generated, and parameter estimation was performed on each sample.

### 4.1 Pearson Type I with concave density

For each of the 100 random samples, with sizes ranging from 100 to 8000, parameters were estimated using the method of moments, maximum likelihood, and Bayesian methods. Figure 3 displays the mean of the 100 parameter estimates for the upper bound $\psi$ as a function of the sample size, as well as the 95% empirical confidence interval. For the Bayesian method, both a Gibbs sampling scheme and the No-U-Turn Sampler (NUTS) algorithm were implemented, yielding similar results.

The three estimation procedures perform very well in estimating the upper bound, which is the parameter of interest in this paper. The mean estimate hovers around the true value of 50, and the confidence intervals include the true value. Estimation remains accurate even for relatively small sample sizes of 2000, which corresponds to approximately 20 years of precipitation data. However, the methods based on likelihood yield more precise results than the method of moments.

### 4.2 Pearson Type I with convex density

Figure 2 shows the mean and the 95% empirical confidence intervals for the samples generated from the Pearson Type I distribution with a convex density. For the method of moments, the estimation of the upper bound is close to the true value of 50. The confidence intervals, wider compared to those associated with the concave density, include the true value. However, estimation variability is very large. It is very sensitive to the sample. For example, for moderate sample sizes around 4000, the

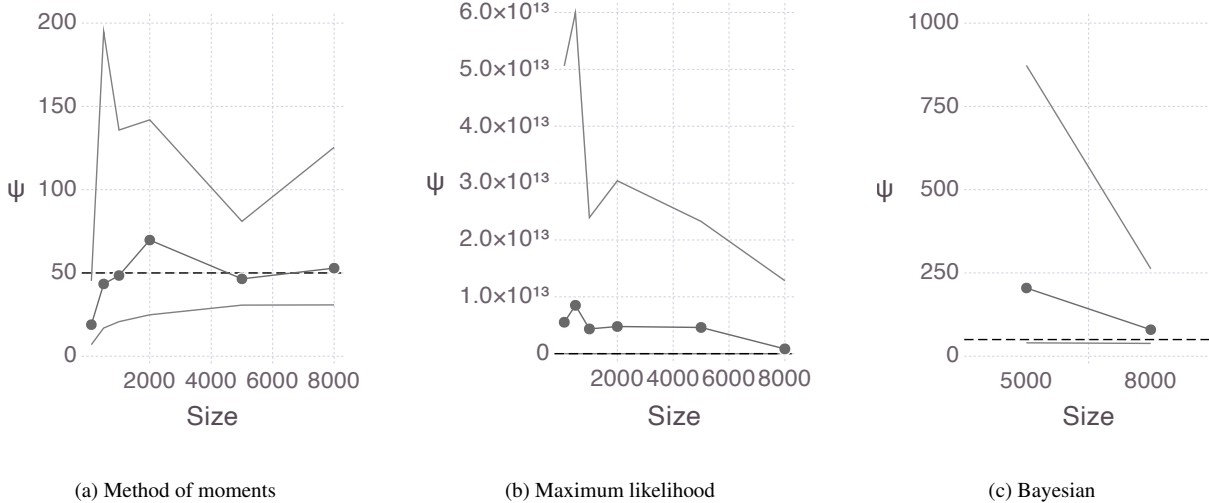

(a) Method of moments        (b) Maximum likelihood        (c) Bayesian

**Figure 4.** Mean and 95% empirical confidence interval for the $\psi$ estimates of the 100 samples of the $Pearson Type 1(0, 50, 1/10, 6)$ distribution obtained with (a) the method of moments, (b) the maximum likelihood and (c) the Bayesian methods using Gibbs sampling.

upper bound estimate average is around 50, but for some samples, the estimate exceeds 100, which is two times larger than the true value. For other samples, the estimate is smaller than 25, which is half the true value.

Upper bound estimates using the maximum likelihood and Bayesian methods are not useful, as shown in Figure 2. The non-identifiability issue arises because the shape parameter $\beta$ compensates for the larger upper bound. While an informative prior for the upper bound could be introduced to control this issue, it would need to be highly informative. However, this approach was not pursued because using such a restrictive prior defeats the purpose of removing subjectivity in PMP estimation.

The sensitivity to the sample and the non-identifiability issue are resolved with very large sample sizes as shown in Figure 5. The estimates are well stabilize around a sample size of 40,000. For precipitation in Canada, it corresponds to approximately 400 years of data.

## 4.3 Key findings from the simulation study

For the Pearson Type I distribution with a concave density, parameter estimates are precise with all three estimation methods considered. For a convex density, the non-identifiability issue in the likelihood is too severe to obtain usable upper bound estimates for common sample sizes using the maximum likelihood and Bayesian methods. However, these two methods could be used with very large sample sizes.

Although the method of moments is sensitive to the data, it provides realistic estimates of the upper bound, even for common sample sizes. This method should be favored for parameter estimation of non-concave Pearson Type I distributions.

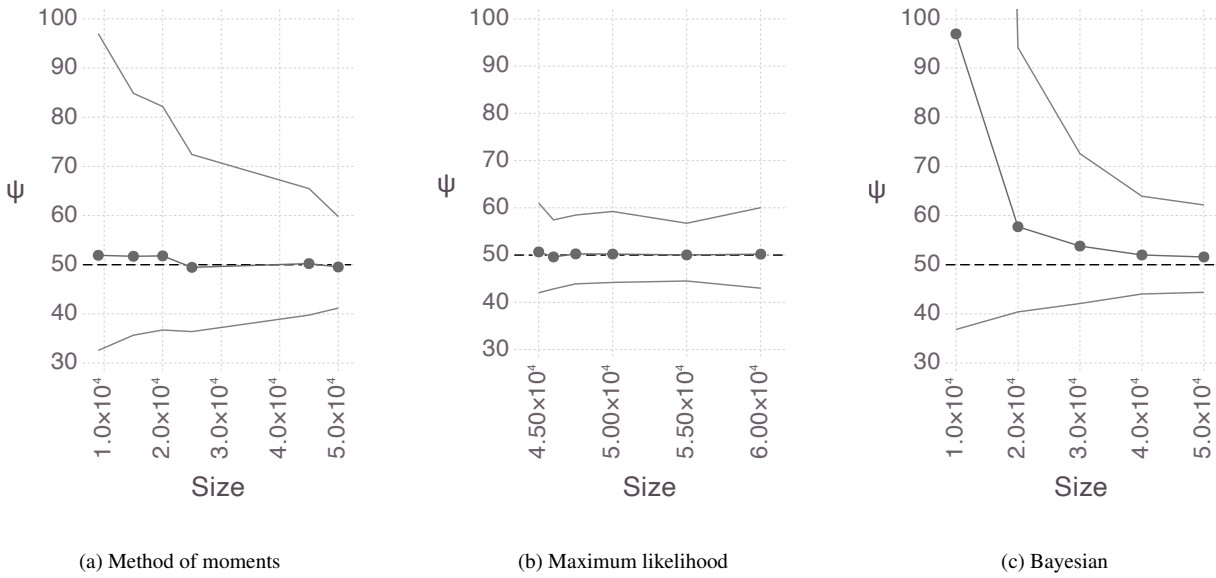

(a) Method of moments      (b) Maximum likelihood      (c) Bayesian

**Figure 5.** Mean and 95% empirical confidence interval for the $\psi$ estimates of the 100 very large samples of the $PearsonType1(0, 50, 1/10, 6)$ distribution obtained with (a) the method of moments, (b) the maximum likelihood and (c) the Bayesian methods using Gibbs sampling.

## 5    Probable maximum precipitation estimation

The Pearson Type I distribution is fitted to the non-zero precipitation data recorded at Montréal and St-Hubert, with the upper
bound estimate assumed to represent the PMP. As shown in Figure 2, the non-zero precipitation density appears to be convex,
so the model is fitted using the method of moments.

### 5.1    PMP estimation for Montréal

The Pearson Type I distribution has been fitted to the 5321 non-zero daily summer precipitation with the method of moments.
The parameter estimates are as follows:

$$\hat{\psi} = 270.0 \quad (141.6, 938.9)$$
$$\hat{\alpha} = 0.4577 \quad (0.3881, 0.5349)$$
$$\hat{\beta} = 18.81 \quad (9.014, 71.99)$$

The values in parentheses correspond to the 95% confidence intervals estimated by non-parametric bootstrap using 10,000
samples. Figure 6 shows the upper bound estimate for each bootstrap sample. The uncertainties of the second shape parameter
$\beta$ and the upper bound $\psi$ are very large, which is expected given the simulation study results for a convex density. Note that
using the maximum likelihood and Bayesian methods does not yield valid estimates due to identifiability issues.

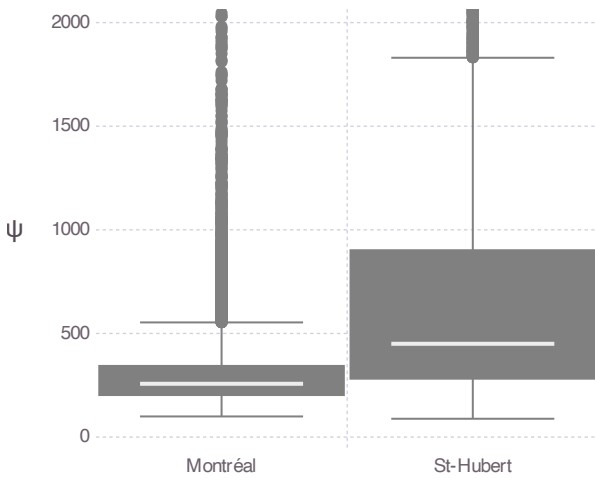

**Figure 6.** PMP estimates (in mm) obtained by non-parametric bootstrap at Montréal (QC) and St-Hubert (QC).

Figure 7a shows the Pearson Type I distribution fitted to the Montréal data. The model fits the data very well. The PMP estimate obtained from the fitted Pearson Type I distribution is consistent with the estimate derived using the moisture maximization method based on Eq. (1). The former estimates the PMP at 270 mm, while the latter estimates it at 284 mm. Unlike 315 the proposed method, the moisture maximization method does not provide an uncertainty estimation.

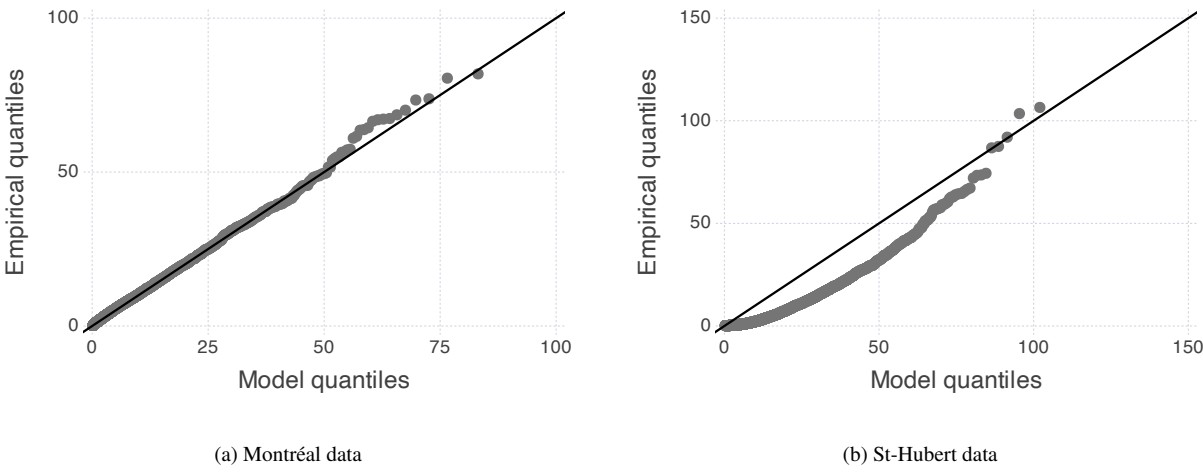

(a) Montréal data                                    (b) St-Hubert data

**Figure 7.** QQ-plots of the fitted Pearson Type I model for (a) the Montréal data and (b) the St-Hubert data.

## 5.2 PMP estimation for St-Hubert

For St-Hubert, the Pearson Type I distribution has been fitted to the 5303 non-zero daily summer precipitation events, and the parameter estimates obtained with the method of moments are as follows:

$$\hat{\psi} = 416.5 \quad (165.0 \, , 9006)$$

$$\hat{\alpha} = 1.463 \quad (0.4381 \, , 1.566)$$

$$\hat{\beta} = 34.75 \quad (15.98 \, , 645.9)$$

where the values in parentheses correspond to the 95% confidence intervals estimated by non-parametric bootstrap using 10,000 samples. Uncertainties in the upper bound and the second shape parameter are exceedingly high, indicating that the non-identifiability issue is more pronounced for these data. Figure 6 shows the upper bound estimates for each bootstrap sample.

Figure 7b shows that the model does not fit the St-Hubert data well. The PMP estimate given by the fitted Pearson Type I distribution (417 mm) is consistent with the estimate obtained using the moisture maximization method (436 mm). This highlights the importance of using a statistical method, which allows for an assessment of estimation quality.

As with the Montréal data, maximum likelihood and Bayesian methods do not yield valid parameter estimates. It should be noted that the estimate for the first shape parameter $\alpha$ with the method of moments is larger than 1, which is inconsistent with the convex form of the distribution. However, the confidence interval includes values smaller than 1.

## 6 Discussion

### 6.1 Pros and cons of the proposed approach

The proposed statistical approach to estimate the PMP translates the usual definition of the PMP into a statistical distribution for the recorded precipitation. The PMP constitutes one of the three parameters, and the remaining two concern the shape of the distribution. By estimating the parameters using standard statistical approaches, such as the moment, maximum likelihood, and Bayesian methods, it is possible to adequately describe the uncertainty, particularly for the PMP parameter. Additionally, the proposed approach uses all the precipitation recorded at the station rather than only a subset from the stations and neighbouring stations. This reduces the subjectivity present in standard approaches.

The major drawback of the proposed approach lies in the non-identifiability issue when the data distribution is convex, as is the case for precipitation. Maximum likelihood and Bayesian methods become very unstable, and this issue also affects the method of moments, although to a lesser extent. Regularized maximum likelihood or informative priors could be used to address the non-identifiability, but the constraints that have to be added to control it are quite narrow. We felt that this added too much subjectivity to the proposed approach and that it would lose its benefits compared to the standard PMP estimation approaches.

In the simulation study, it is shown that the non-identifiability issue vanished with very large sample sizes. Figure 5b shows that the maximum likelihood estimation is stable from a sample size of 45,000. If we consider that 100 storms occurs during a year, such sample size would correspond to 450 years of observation. Of course, no meteorological record is that long, but it could be possible to have such a sample size of synthetic storms generated with a storm generator. However, such data augmentation should be carefully implemented to avoid overconfidence. For instance, if 40,000 daily precipitation data points generated from a weather generator are used to estimate the model, do these 40,000 data points contain 400 times more information than a actual recorded series of size 100? At this point, this is beyond the scope of the present paper, but it could be an interesting avenue for future investigation.

Another alternative to increase the sample size would be to include information from nearby stations. This could be achieved within the Bayesian framework described in Section 3.2.3 by replacing the parameter prior distribution with a spatial prior. However, the dependence between stations would need to be modeled in the likelihood, as a single storm can generate precipitation across multiple stations. Accounting for this dependence would decrease the effective sample size of the pooled stations, and we believe that this effective sample size might not reach the level where the estimates are stable. However, we could be wrong.

PMP estimation, whether with the proposed method in this paper or with more standard approaches, is very sensitive to the data due to non-identifiability. For the two nearby meteorological stations considered, i.e., the Montréal Pierre-Elliott-Trudeau International Airport and the St-Hubert Airport stations located 26 km apart, the PMP estimates are very different. However, these two stations do not experience significantly different climates and storms. Furthermore, in several estimates provided by engineering consulting firms, storms from even more distant stations are combined to estimate the PMP, a practice known as storm transposition. Moreover, the extreme-value analysis of the precipitation at these two stations, presented later in Section 6.2, yields consistent return level estimates. Therefore, the difference in the PMP for these two stations is more a numerical problem related to the PMP definition than a genuine difference in the PMP.

Another drawback is fitting the proposed short-tailed statistical model to heavy-tailed precipitation data as shown in the following section. This inconsistency could explain why the model does not fit the St-Hubert data well, as shown in Figure 7b.

## 6.2  Comparison with extreme value analysis

For the purpose of comparison, an extreme value analysis have been performed on the precipitation data of Montréal and St-Hubert. The Peaks-Over-Threshold (POT, Davison and Smith, 1990) extreme-value model has been fitted by maximum likelihood to the Montréal data, with the threshold of 30 mm chosen using the mean residual life plot method as described by Coles (2001, Chap. 4). The estimated parameters of the generalized Pareto distribution modelling the excesses above the threshold are as follows:

$\hat{\sigma} = 9.95 \quad (7.96, 12.45)$

$\hat{\xi} = 0.0421 \quad (-0.1288, 0.2131)$

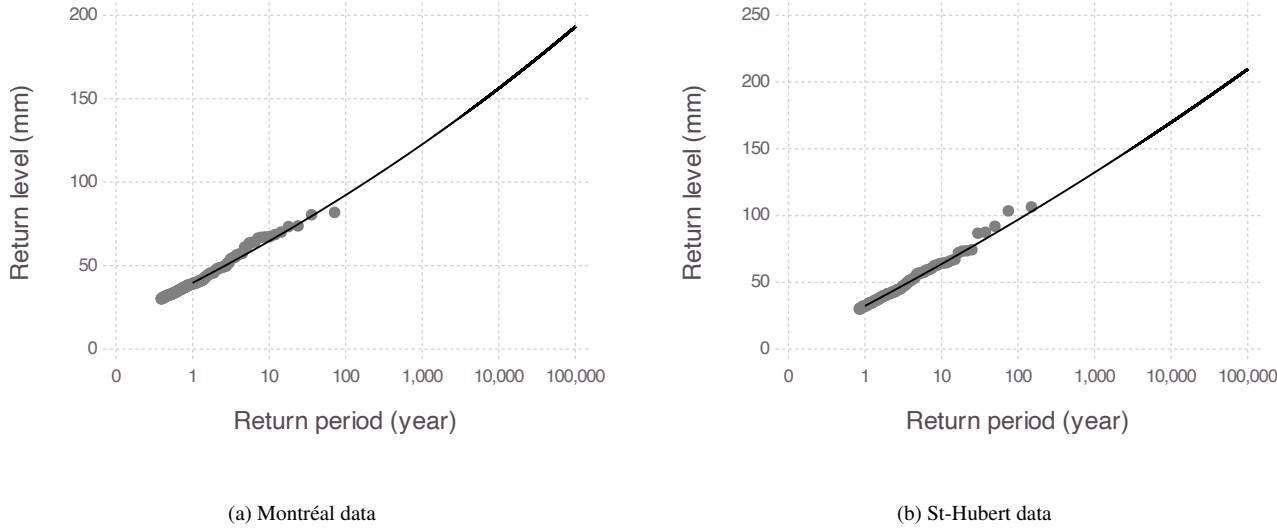

(a) Montréal data
(b) St-Hubert data

**Figure 8.** Return level plot of the fitted Peaks-Over-Threshold model for (a) the Montréal data and (b) the St-Hubert data.

where the values in parentheses correspond to the 95% confidence intervals estimated using the Fisher information matrix. The model fits the data very well, as shown by the return level plot in Figure 8a. Note that the shape parameter estimate is positive, indicating an unbounded heavy-tailed distribution, which is typical for precipitation but inconsistent with the PMP existence assumption. Nevertheless, a short-tailed distribution cannot be excluded, as the shape parameter confidence interval includes negative values. As an indication, the 10,000-year return level estimated with the POT model is 156 mm ($58\ mm$, $255\ mm$), and the PMP value of $270\ mm$ corresponds to a return period longer than 10 million years.

The POT model has also been fitted to the St-Hubert data. Parameter estimates are as follows:

$\hat{\sigma} = 13.1594 \quad (10.5917, 16.3494)$

$\hat{\xi} = 0.0259 \quad (-0.1335, 0.1854);$

and Figure 8b shows the model fit to the data. Again, the model fits the data very well, and the shape estimate is positive, indicating a heavy-tailed distribution. Using the fitted POT model, the 10,000-year return level estimate is 170 $mm$ ($80\ mm$, $259\ mm$), which is consistent with the corresponding estimate of $156\ mm$ for Montréal, located $26\ km$ apart. The PMP estimate of $416\ mm$ corresponds to a return period longer than 1 billion years.

### 6.3 Possible modifications of the statistical model

To reduce the impact of model non-identifiability, we also developed a new parametrization for the Pearson Type I distribution, replacing the shape parameters with a location parameter $\mu > 0$ and a concentration parameter $\nu > 0$:

$$\mu = \frac{\alpha\psi}{\alpha + \beta} \quad \text{and} \quad \nu = \alpha + \beta.$$

We therefore have $\alpha = \frac{\mu\nu}{\psi}$ and $\beta = \frac{\nu}{\psi}(\psi - \mu)$. However, even with this parametrization, non-identifiability remained an issue for maximum likelihood and Bayesian inference.

Another approach to imposing a short-tailed model compliant with the PMP concept would be to consider the reverse-Weibull distribution. The reverse-Weibull is obtained by imposing a negative shape parameter on an extreme value distribution. While this choice results in a short-tailed distribution, it is difficult to justify this constraint beyond the fact that it produces

an upper bound. Moreover, we are concerned that using an extreme value distribution in an inappropriate context, such as by imposing a negative shape parameter when the data suggests a positive one, could give practitioners a false sense of security. They might believe they are operating within the extreme value framework when they are not.

### 6.4 Non-stationarity

For the observed data considered, there is no evidence of a trend in either the precipitable water or precipitation, as shown in

Figure 1. Non-stationarity might be present in long series of simulated data from a climate model. In such cases, the proposed statistical model could easily be extended to account for non-stationarity, and also for seasonality if needed. Eq. 6 could be generalized to incorporate seasonality and non-stationarity by allowing either or both the PMP and the shape parameters to evolve over time. For example, precipitation $Y_{tsi}$ of year $t$, season $s$, and event $i$ can be modeled as a function of the year and the event $i$ as follows:

$$Y_{tsi} \sim PearsonType1(0, \psi_{ts}, \alpha_{ts}, \beta_{ts});$$

where the year $t$ and season $s$ could serve as covariates.

### 6.5 Recommendations

Although translating the definition of the PMP into a statistical model is interesting, and despite the possibility of including non-stationarity and estimating uncertainty, we do not recommend using the Pearson Type I distribution to estimate the PMP. The

non-identifiability makes the model too sensitive to the data, and the PMP estimate becomes too volatile. This problem is also present in the standard moisture maximization method. Therefore, we align with the conclusions of the National Academies of Sciences, Engineering, and Medicine (2024), which recommends an extreme value analysis instead. Additionally, the extreme-value theory allows for a genuine estimation of the uncertainty of extreme values, even when extrapolating to return periods that exceed the range of the data. Furthermore, it is easily generalizable to non-stationary cases, allowing the integration of the

effects of climate change.

More generally, the fact that PMP estimates using either moisture maximization, Hershfield's method, or the Pearson Type I method are so sensitive to the data is a critical concern from an engineering standpoint. Specifically, for the Pearson Type I method with a convex density, depending on the data, the PMP estimate can range from half to more than twice the true value. In the former case, using the estimate would result in under-dimensioning the infrastructure, putting the public at risk.

In the latter case, using it would result in over-dimensioning the infrastructure, thereby increasing costs and environmental impacts. Although uncertainty estimates are not available with the moisture maximization and Hershfield's methods, the fact

that the corresponding PMP estimates for Montréal and St-Hubert were so different is an important indication of the methods' sensitivity.

In the case of this article, we have seen that the POT model fits the data from both stations very well and that the estimates of the 60,000-year precipitation were consistent. Moreover, the extreme value analysis indicates an unbounded and heavy-tailed distribution of precipitation, which is consistent with numerous results in the literature (e.g. Papalexiou and Koutsoyiannis, 2013). Therefore, it is better to design infrastructure by setting an appropriate level of risk and evaluating the uncertainty of the estimate.

## 7    Conclusions

In this study, we developed a new statistical model for estimating the PMP based on its definition. The model involves modelling daily precipitation with the Pearson Type I distribution, where the upper bound corresponds to the PMP. As a proper statistical model, parameter and uncertainty estimations can be derived using well-known statistical methods.

Our analysis demonstrates that while the proposed statistical approach offers potential benefits—such as translating the PMP definition into a statistical model, incorporating non-stationarity, and providing uncertainty estimates—significant drawbacks limit its practical application. The major challenge lies in the non-identifiability issue, which renders the model highly sensitive to data and leads to volatile PMP estimates. This issue persists despite attempts at reparametrization and the use of regularized maximum likelihood or informative priors, which introduce subjectivity that undermines the model's advantages.

Given the inherent challenges and limitations of the Pearson Type I distribution for precipitation modelling, we recommend using extreme value analysis for PMP estimation. This approach aligns with the findings of the National Academies of Sciences, Engineering, and Medicine (2024), which advocate for extreme value analysis due to its robustness and applicability, even in the context of non-stationary conditions brought about by climate change. With our data, the 60,000-year return level estimates of daily precipitation at the two considered locations were consistent, in contrast to the PMP estimates for those two locations. Moreover, the extreme value analysis indicated a heavy-tailed distribution, consistent with existing literature, which invalidates the concept of PMP.

Future work may involve estimating the PMP of storms instead of daily precipitation. In this paper, we estimated the daily PMP, but precipitation accumulation over several days could also be of interest. However, accumulation over several days would decrease the sample size and exacerbate the non-identifiability issue. Future work may also focus on PMP estimation based on a large sample of synthetic storms provided by a storm generator.

In conclusion, while innovative statistical methods offer promising avenues for PMP estimation, traditional extreme value analysis remains, in our opinion, the most practical and reliable approach for assessing precipitation extremes and guiding infrastructure design.

*Code and data availability.* The data and code for reproducing the results are provided in the public repository: https://github.com/JuliaExtremes/PMP.jl.

*Author contributions.* Authors' Contribution statement using CrediT with degree of contribution:

**Anne Martin:** Formal Analysis (lead), Investigation (lead), Methodology (lead), Software (equal), Validation (equal), Visualization (equal), Writing – Original Draft Preparation (equal), Writing – Review & Editing (equal).

   **Élyse Fournier:** Conceptualization (equal), Funding Acquisition (equal), Investigation (supporting), Methodology (supporting), Supervision (supporting), Writing – Original Draft Preparation (supporting), Writing – Review & Editing (supporting).

   **Jonathan Jalbert:** Conceptualization (equal), Funding Acquisition (equal), Investigation (supporting), Methodology (supporting), Software
(equal), Validation (equal), Visualization (equal), Supervision (lead), Writing – Original Draft Preparation (equal), , Writing – Review & Editing (equal).

For more information, please see the taxonomy website.

*Competing interests.* The is no competing interest to declare.

*Acknowledgements.* This work was supported by Natural Sciences and Engineering Research Council of Canada, Hydro-Québec, MITACS
Acceleration program and the ARRIMÉ research alliance. We would like to thank Gabriel Gobeil (Environment and Climate Change Canada) for his valuable assistance, as well as Julie Carreau and Jean-Luc Martel for their insights.

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
