# Peer review of "Statistical estimation of probable maximum precipitation"

_EGUsphere, 2024_

## Author Comment (AC1)

**EGUSPHERE-2024-2594**
**Detailed Responses to Reviewer 1's Comments**

Anne Martin, Élyse Fournier and Jonathan Jalbert

December 20, 2024

First, we would like to thank the three reviewers for their thorough review and relevant comments and suggestions. Based on their feedback and recommendations, we notably adjusted the structure of the paper and added numerous clarifications. The modifications in the revised version of the manuscript are highlighted in blue. Below are detailed responses to all of Reviewer 1's comments.

I agree with the authors when they state that "translating the definition of PMP into a statistical model is interesting" (line 350). They could also say "estimating PMP is a really hard problem". To the authors' credit, they begin with the commonly accepted definition of PMP as an upper bound, and then construct a statistical model which fits this definition. Fitting a model with a finite upper bound is challenging because precipitation data usually suggests that the distribution is unbounded, and further has a heavy tail. It is probably not surprising that the authors ultimately find their approach to be unsuitable for implementing in practice, and conclude that the best statistical approach is to eschew the upper bound requirement and instead implement extreme value (EV) methods.

Thank you. This is a very good description of the proposed contribution.

**Major concerns**

1. Unfortunately, I think the manuscript's structure does not tell its story well. Primarily I view the paper as an interesting way to discuss the challenges of PMP estimation, and talking about their particular model is one part of this larger story. It strikes me that the take away message does not appear in the abstract or in the body until Section 6. I think it would be better to move these messages up front. The story I imagine is something like this:

   (i) Statistically estimating PMP is hard because its definition assumes a bounded tail, but precipitation data suggests the tail is unbounded. Because statistical estimation is hard, other methods like moisture maximization and Herschfield's scaling get used. Uncertainty and climate change are hard to incorporate into these

non-statistical methods and frequently-used moisture maximization approaches involve several subjective judgements.

(ii) Starting with the ideas which underlie moisture maximization, we develop a sensible statistical model which assumes an upper bound.

(iii) We perform simulation studies and use the method to fit PMP at two locations in Quebec, but find that estimates for the upper bound have unsuitable uncertainty.

(iv) We conclude with a discussion and offer our suggestion for best practices.

I think all the pieces of this story are in the paper, but I do not think the current focus of the paper gets the essential message across very well.

This is true, and we fully agree with your suggestion. In the revised version of the manuscript, the sections have been rearranged as you suggested to more effectively convey the conclusions.

2. I find the notation in the paper to be inconsistent. In Equation (1), $Y_i$ denotes precipitation of storm $i$, but in Equation (2) I believe $Y_i$ has been replaced by $P_i$. Equation (4) supposedly comes from Eq. (1), but has quantities $EP_i$ and $EP_{max}$, which are presumably $PW_i$ and $PW_{max}$ in Equation (1)?

This was a mistake; thank you for pointing it out. In the revised manuscript, we consistently used "PW" for precipitable water. "EP" was the equivalent acronym used in the French version.

3. The ratio $EP_i/EP_{max}$ is known/assumed to be less than 1, correct? If so, please say this explicitly.

Yes, it is assumed to be less than or equal to 1. This is now explicitly mentioned in the revised version.

4. I believe Equation (6) is used as the basis for the statistical model: $Y_i = EP_i/EP_{max} \times r_i \times PMP$. If I am following correctly, $Y_i$ is random and observed. I think $EP_i$ and $r_i$ which underlie $Y_i$ are random, but unobserved. $EP_{max}$ is a parameter but not known, and PMP is the parameter we wish to estimate. So in the end, the authors propose a model for the observed precipitation $Y_i$, but use moisture maximization logic to include PMP as a parameter. They choose a beta/Pearson 1 as their distribution to fit. A cynical comment could be "the authors use a data-independent argument to conclude the data arise from a distribution, but which fits the data poorly". I think the story to be told here is that if one begins with a supposition of an upper tail, and one tries to then fit a model based on that assumption, things are really hard.

Exactly. We indeed "propose a model for the observed precipitation $Y_i$, but use moisture maximization logic to include PMP as a parameter". Again, we agree with your comment that if one assumes an upper bound, things become quite challenging. We wanted to show that the issue might not lie with the Pearson Type I model itself but with the hypothesis of PMP existing as the upper bound. Elements of this discussion have been added to the revised version.

5. If I understand correctly, the authors propose a beta/Pearson 1 distribution and fit **all** of the nonzero rainfall data to it. There is talk of thresholding on page 8, but it seems to be more tied to the discrete nature of the measurements rather than to thresholding for focusing on extremes.

   Exactly. The measurement precision is 0.1 mm with the lowest non-zero value of 0.2 mm. With the thresholding, we wanted to assess whether this discretization, which has a greater impact on small precipitation amounts, would affect the overall fit of the model. It turns out that the discretization does not have a noticeable effect on the overall fit.

6. An EV approach would pick a high threshold or take block maxima and fit an EV model, presumably a reverse-Weibull guaranteeing an upper bound. Would such a method be better suited for estimating an upper bound than fitting a beta to the entire distribution?

   Thank for the suggestion. As mentioned in a previous comment, the goal was to propose a statistical model for the PMP based on the moisture maximization logic. While it is true that imposing a negative shape parameter on an extreme value distribution will result in an upper bound, this choice is difficult to justify beyond the fact that it produces an upper bound. Moreover, we are concerned that using an extreme value distribution in an inappropriate context, such as by imposing a negative shape parameter, could give practitioners a false sense of security. They might believe they are operating within the extreme value framework when they are not. Elements of this discussion have been added to the revised version of the manuscript.

7. The authors show QQ plots for the EV models in Figure 6. QQ plots for the beta fit are noticeably absent.

   QQ plots of the Pearson Type I fit are provided in Section 5 of the revised manuscript. Thank you for the suggestion.

8. l173. Why is $\beta$ known to be greater than 1?

   Typically, precipitation has a monotonic decreasing density. This behavior is achieved with the beta distribution when $\alpha < 1$ and $\beta > 1$. This clarification has been included in the revised version.

9. l304: Figure??

   Thank you for pointing that out.

---

## Author Comment (AC2)

**EGUSPHERE-2024-2594**
**Detailed Responses to Reviewer 2's Comments**

Anne Martin, Élyse Fournier and Jonathan Jalbert

December 20, 2024

First, we would like to thank the three reviewers for their thorough review and relevant comments and suggestions. Based on their feedback and recommendations, we notably adjusted the structure of the paper and added numerous clarifications. The modifications in the revised version of the manuscript are highlighted in blue. Below are detailed responses to all of Reviewer 2's comments.

The paper under consideration presents a statistical approach using the Pearson Type I distribution to estimate the upper bound of historical rainfall, incorporating uncertainty bounds. The authors aim to quantify uncertainty and address subjectivity inherent in the various stages of the World Meteorological Organization (WMO)-recommended Probable Maximum Precipitation (PMP) estimation methods. However, the WMO-recommended moisture maximization method focuses on maximizing highly efficient storms based on the physical mechanisms of those storms. From a statistical perspective, precipitation's tendency to exhibit a heavy-tailed distribution poses challenges in defining an upper bound, and this study similarly encounters this issue.

**Major concerns**

1. Previous studies demonstrated that precipitation naturally often exhibits a heavy-tailed distribution (shape parameter greater than 0) that brings rare storms over global scale, the proposed method struggled to estimate upper bound of those places by majority of parameter estimation methods. In this study, the method of moments partially quantifies the range with limited data, but the range is unrealistically large, for example, the range estimated for St-Hubert station varies between 165 to 9006.

   It is also the case here that precipitation is heavy-tailed and unbounded, as shown through the classical extreme value analysis in Section 6.3. In this paper, we proposed a statistical model for the PMP with a finite upper bound based on the moisture maximization logic. However, since this model does not fit the data well and is highly sensitive to it, as shown by the non-parametric bootstrap confidence intervals, it suggests that the PMP definition based on the moisture maximization logic may not be appropriate. We therefore recommend using extreme value theory instead.

2. A simulation study is conducted with distribution assumed convex density and found more than 40,000 sample size (in arid/semi-arid region that equivalent to more than 1000 years wet days) is required to stabilize the estimate. Given this, it is surprising that the authors did not attempt to expand the sample size for the two stations by incorporating numerical model ensemble precipitation products. Doing so could have supported their findings.

   This is a good suggestion to augment the actual precipitation data with simulated precipitation from a climate model or a weather generator. It is indeed something we could consider if we were to use this proposed model to estimate the PMP, although this is not our final recommendation in this paper. However, such data augmentation should be carefully implemented to avoid overconfidence. For instance, if 40,000 daily precipitation data points generated from a weather generator are used to estimate the model, do these 40,000 data points contain 400 times more information than a recorded series of size 100? At this point, this is beyond the scope of the present paper, but it could be an interesting avenue for future investigation. Elements of this discussion has been added in the Discussion of the revised version.

3. This study compares their estimated upper limit with moisture maximization based PMP value. The PMP values using moisture maximization were found to be 282 mm for Montréal and 436 mm for St-Hubert, whereas the observed 24-hour maximum precipitation for these stations was 81.9 mm and 106.5 mm, respectively. Thus, the maximization ratio will be 3.44 and 4.09. The reason behind the exceptionally high maximization ratio may be due to the selection of storms and/or estimation of storm associated precipitable water. This study includes low magnitude storms (0.9 quantile might give more than 500 samples but previous studies mostly consider the highest 50 or less storms) and did not separate those storms that could lead to higher maximization ratio. Previous studies mostly limit the maximization ratio 2.0 (that only for orographic storms). Imposing a similar limit could provide some physically possible value around 200mm that aligns with 10,000-year return level (POT based) and PMP value would not much different within 26 km distance. Since the moisture maximization method provides an unrealistically high PMP value, comparing with this value to validate the method is questionable. It is recommended to use multiple study sites and consider those sites where maximization ratio lies below 2.0 and compare within those sites.

   PMP estimates using the moisture maximization method in Section 2.2 are based on the top 10% of storms, as suggested by Clavet-Gaumont *et al.* (2017). The reference has been added in the revised version. A sensitivity analysis was performed, and regardless of the quantile used for storm selection (10%, 1%, or 0.1%), the PMP result remained unchanged. This is because the event of September 20, 1989, with 63.8 mm at the Montreal station, is maximized regardless of the storm threshold. The same applies to the St-Hubert station, where the precipitation of 73.4 mm on July 5, 1958, is maximized. The maximization ratios are 4.4 and 4.9, respectively. Since precipitable water was not directly observed, it was estimated using the dew point, which may indeed affect the quality of the PMP estimates.

The approach you suggested to improve PMP estimates using the usual moisture maximization method would be appropriate if that were the goal of the paper. However, we applied the simple moisture maximization method with the usual formulas at these stations primarily to highlight, to some extent, the flaws of the methodology and the need for great care in obtaining sensible PMP estimates. While the approach you recommend, choosing sites where the maximization ratio is below 2.0, may work for those specific sites, it would not be directly replicable to other locations. Therefore, although the elements you propose to improve the PMP estimates have been added to the manuscript, we believe this is beyond the scope of the present paper, as our focus is on the developed model for statistical PMP estimation.

4. National Academies of Sciences, Engineering, and Medicine (2024) recommends for risk-informed extreme value analysis methods that account for low exceedance probabilities and provide robust uncertainty and nonstationary quantification. It remains unclear how the proposed method offers advantages over or resolves issues better than these recommended approaches.

   In this paper, we provide additional support for using extreme value theory, as recommended by the National Academies of Sciences, Engineering, and Medicine (2024). The newly proposed statistical approach for PMP estimation does not perform well, as it requires an extensive amount of data and is highly sensitive to the data. In the revised version of the manuscript, we emphasized more clearly that we align with the conclusions of the National Academies of Sciences, Engineering, and Medicine.

5. The choice of Pearson Type I distribution over other distributions is missing.

   We have added more details in Section 3.1 explaining why the Pearson Type I distribution is the natural choice for the moisture maximization logic.

**Minor comment:**

- Line 304: The placeholder "Figure??" needs Figure number.

  Thank you for pointing that out. It is now corrected.

- Additionally, there is inconsistent notation in Equation 4 compared to Equation 1.

  Yes indeed, thank you.

- The term "EP" and "EPmax" should be clearly defined to maintain consistency and avoid confusion.

  This is a translation mistake from our part. Thank you for pointing that out. It is now corrected.

---

## Author Comment (AC3)

**EGUSPHERE-2024-2594**
**Detailed Responses to Reviewer 3's Comments**

Anne Martin, Élyse Fournier and Jonathan Jalbert

December 20, 2024

First, we would like to thank the three reviewers for their thorough review and relevant comments and suggestions. Based on their feedback and recommendations, we notably adjusted the structure of the paper and added numerous clarifications. The modifications in the revised version of the manuscript are highlighted in blue. Below are detailed responses to all of Reviewer 3's comments.

Interesting paper. However, I feel it may become more of use with additional investigations of refinements needed to the method that has been proposed. See my comments below.

1. l125: Is $EP_{\max}$ defined for a calendar day for nearby regions or defined as the maximum at that point location? Equation 1 referred to $PW_{\max}$ which is the atmospheric moisture. $EP_{\max}$ is I suspect the same but should be consistent or else defined. If non-seasonal, please refer to WMO guidelines for seasonal variations in PMP.

   Precipitable water should be written as $PW_{\max}$ and not $EP_{\max}$ throughout the paper; this has now been corrected. To answer your question, $PW_{\max}$ is defined as the maximum at that specific location. This has been clarified in the revised version, and we have also referred to the WMO guidelines regarding seasonal variations. Thank you for the suggestion.

2. l140 - interesting. However, the sampled $r_i$ are non-iid, which complicates their use in defining the Beta distribution I think. Plus, there is an assumption that the sampled $r_i$ has an upper limit of 1. Given this limiting value will dictate/influence the PMP estimate, the uncertainty associated with this assumption is important to characterize.

   It is true that the sampled $r_i$ are not iid. Typically, for precipitation in the considered location, autocorrelation exists in daily non-zero precipitation series, but it is very weak and short-range. We believe that the impact of this very small dependence is quite limited compared to the overall sampling uncertainty, which already results in very large uncertainty in the parameter estimates. For example, the autocorrelation estimate for the non-zero precipitation series recorded in Montréal is 0.0092 for a lag of one day, and 0.0095 for the St-Hubert station. These values have been added to Table 1 to demonstrate that dependence is weak for the data considered. However, in regions where autocorrelation is stronger, it should be accounted for.

In this paper, the analysis is restricted to summer precipitation (from May to October inclusive) to minimize the effects of seasonality. While some seasonality may still exist, it appears negligible compared to the natural variability of precipitation and precipitable water as shown in Figure 1. We have added these figures to the revised version of the manuscript.

3. Also, how can stationarity be assumed given there is a clear temporal trend in precipitable water time series. Would violation of the stationarity assumption distort the beta distribution parameters?

   Yes, non-stationarity in precipitable water and/or in precipitation would distort the Pearson Type I distribution. For the considered observed data, there is no evidence of a trend in either the precipitable water and precipitation, as shown in Figure 1. This is due to the natural variability of these variables.

   Non-stationarity might be present in long series of simulated data from a climate model. In such cases, Eq. (6) should be extended to account for non-stationarity by allowing either or both the PMP and the ratio to evolve over time. In the context of the Pearson Type I distribution, this involves allowing the upper bound and the shape parameters to vary with time. This discussion has been added to the revised version of the manuscript.

4. l375 - The authors recommendation makes sense. In addition to the issues they have mentioned, I also feel that the lack of independence (unless they parameters are being fitted using iid data above a threshold) and the presence of a trend are limiting factors. I wonder if using a nonstationary model and regional data can help overcome these limitations.

   Thank you. We agree that a regional model could potentially help in the estimation of the PMP. Such an approach would involve modelling the spatial dependence between precipitation at multiple sites. In future work, we plan instead to focus on spatial modelling of extreme precipitation within the framework of extreme value theory.

[Figure]

(a) precipitation            (b) precipitable water

Figure 1: Time series of (a) daily precipitation and (b) precipitable water for the top 10% of storms recorded at the Montréal station.

---

## Referee Report (RR1)

I appreciate the authors' efforts in revising the manuscript, particularly their acknowledgment of the limitations of their proposed method for practical applications and their alignment with the National Academies of Sciences, Engineering, and Medicine (NASEM, 2024) recommendation to use an Extreme Value Theory (EVT)-based approach. However, the study's primary contribution lies more in its critical evaluation of PMP methodologies rather than in the statistical model itself. While previous studies have highlighted the flaws in the WMO-recommended PMP estimation methods, this study uniquely examines the limitations of physically based moisture maximization through a statistical framework before ultimately recommending an EVT-based approach.

That said, I still have few concerns regarding the robustness of the conclusions, primarily due to the limitations of the EVT analysis and the inadequate discussion of sampling uncertainty.

**Concerns:**

- o  The challenge of estimating very long return periods (e.g., 10,000 years) using only 75 years of data remains unresolved. Such extreme quantile estimates require substantial extrapolation, which increases uncertainty. Moreover, the stability of estimation may be sensitive to the threshold selection in the Peaks-Over-Threshold (POT) method. I suggest going over (NASEM, 2024) report about sampling uncertainty. The authors should explicitly discuss these uncertainties in the EVT approach to prevent misinterpretation by end-users. Otherwise, there is a risk of conveying an overconfident message about the reliability of these estimates.

- o  The authors acknowledge that identifiability issues affect the reliability of PMP estimates, yet the discussion remains largely theoretical. The manuscript would benefit from a practical demonstration in section 6.3 of how alternative constraints—such as regularized maximum likelihood estimation or Bayesian priors—influence PMP estimates. If these methods were tested but found ineffective, the authors should clearly articulate why. This would strengthen the argument against using Pearson Type I for PMP estimation.

---

## Referee Report (RR2)

The authors have provided satisfactory responses and made the necessary corrections. The manuscript can be accepted in its current form.

---

## Author Response (AR2)

**EGUSPHERE-2024-2594**
**Detailed Responses to Editor Comments**

Anne Martin, Élyse Fournier and Jonathan Jalbert

May 29, 2025

Specific comments

1. L3: I recommend moving the WMO definition from the abstract to the introduction.
   Done. The definition is now at L20.

2. L16: The subheading "Context" is not necessary; consider removing it.
   Done. Thank you for the suggestion.

3. L17–25: The introduction currently focuses too heavily on dam safety, whereas PMP has broader applications in hydrological risk assessment. As HESS is not specifically a journal for dam safety research, I suggest revising this section to present PMP in a broader hydrological context, with dam safety as one example among others.
   Done. Thank you for the suggestion (L15).

4. L35–37: The relevance of PMF is unclear. If it is not directly used in the model development or not specifically mentioned later in the manuscript, I recommend removing it. On a different, more general comment - the introduction could be revised to be less Canada-specific and more broadly applicable to international readers.

   (a) Done. There is stil one mention of the PMF to understand where the PMP can be used (L19).

   (b) We appreciate the suggestion and understand the value of making the introduction broadly applicable. However, we believe it is important to present the Canadian context in which our study was conducted, as it provides relevant background and motivation. We trust that international readers, with a clear understanding of this context, will be able to adapt the proposed method to their own settings.

5. L58–59: Eq. 2 may not be necessary. You could start discussing the ratio on line 60, referencing Eq. 1, which should suffice for clarity.
   Done. Thank you for the suggestion (L54).

6. Section 1.5: This section might be better integrated into the introduction. If the main point is to emphasize that climate models can be used to estimate changes in PMP (you

are not using climate models later in the paper, just discussing their use), I suggest shortening the section and incorporating it earlier in the text.

We have reduced the emphasis on modeling non-stationarity in the paper's objectives, as it was not required for our study (L114). However, we believe the paragraph on climate models remains important. Even without modeling non-stationarity, several sources cited in that section recommend using simulated data from climate models to address the scarcity of observational data for PMP estimation. We have incorporated elements of this discussion in the revised version of the manuscript (L99).

7. L108: Please clarify the acronym "CC" on first use.
   Thank you for pointing that out.

8. L124–128: Consider removing this text; it does not appear essential.
   Done. Thank you for the suggestion.

9. L128–129: The sentence regarding data availability would be more appropriately placed in the data availability section at the end of the manuscript. Can be removed here.
   Thank you for pointing that out.

10. Structural suggestion: You might consider presenting the proposed PMP estimation model (Section 3) before introducing the case study data (Section 2). Since the model is general, introducing it first may improve the manuscript's logical flow. However, this is a suggestion; you may ignore it and choose to retain the current structure.
    Thank you for the suggestion. The methodology is now the Section 2 and the simulation study appears in the Section 3.

11. Figures 1 and 2: These figures are not "a must" and could be moved to the supplementary material as Figures S1 and S2 (supplementary material in a separate file, not as an appendix).
    Figures 1 and 2 were introduced in response to Reviewer 3's comments. These figures help justify that incorporating non-stationarity is not necessary for the analyzed data, as no trend is visible in either precipitation or precipitable water. While we agree that they are not essential, we believe they still have value and merit inclusion in the manuscript. Please let us know which option you consider preferable.

12. Table 2: This table could be merged with Table 1 to avoid redundancy in presenting station-specific information.
    Both tables are merged into Table 2. Thank you for the suggestion.

13. L159–160: Please provide additional information for the EVT analysis: What threshold was selected? Why not use annual maxima? Which distribution was fitted (e.g., GEV)? Consider adding a brief sensitivity analysis on the threshold and distribution choice.
    Additional information has been provided in Section 4.2, which now presents the EVT results. To summarize, while the block maxima approach could have been used, we adopted the Peaks-Over-Threshold (POT) methodology in accordance with the recommendations of the National Academies of Sciences, Engineering, and Medicine (2024).

The threshold was selected using the mean residual life plot, as suggested by Coles (2001) (L281). In the POT methodology, exceedances above the high threshold are modeled using the Generalized Pareto distribution (L280). We have also added uncertainty estimates for both the parameters (Table 4) and the return levels (L282, L349, L353).

14. Section 3.2.1: The method of moments is well established (see your references to Johnson et al.). Consider moving large parts of this section (or entirely) to the supplementary material unless some parts of it are critical to your main argument.
Thank you for pointing that out. Section 2.2.1 of the revised version has been shortened. We retained the key points of the method of moments as applied to our model, the Pearson Type I distribution, which are:

- the moments have closed-form analytical expressions;
- the third and fourth moments do not depend on the upper bound, i.e., the PMP;
- the estimates have tractable forms.

15. Section 5: Subsections are not strictly necessary here; the results can be presented as a single, continuous narrative.
Done. Thank you for the suggestion.

16. L304–307 and L319–321: The parameter estimates would be more clearly presented in a table.
The parameters estimates for Montréal and St-Hubert can now be found in Table 3. Thank you for the suggestion.

17. Figure 7: Please clarify why quantiles close to 100 are missing in panel (a), and why quantiles exceed 100 in panel (b).
Figure 7 shows the quantile–quantile plots. For St-Hubert, the highest empirical quantile is 106 mm, which is why the axes exceed 100 mm. We chose to use quantile–quantile plots here because they place more emphasis on the tail, in contrast to probability–probability plots, where the axes are limited to the unit interval, emphasizing the bulk of the distribution.

18. L314–315: It is somewhat limiting to present the case study using only one of the three proposed estimation methods you described. You demonstrate them using the synthetic data, but it is better also to demonstrate using real case studies. I suggest including an additional case study, possibly using data from another region (you should not limit yourself to Canada), where all estimation methods are applied and compared. This would provide a more robust demonstration of the model's applicability.
Estimates obtained using maximum likelihood and the Bayesian method have not been included, as doing so would contradict the conclusions of the simulation study. The simulation study demonstrated the poor performance of these two approaches due to a lack of parameter identifiability. For the Montréal data, the PMP maximum likelihood estimate exceeds $2 \times 10^{13}$ mm. Elements of this discussion have been incorporated into the revised version of the manuscript in L368–381, where we address the regularized

likelihood method. The absurd PMP maximum likelihood estimate is specifically discussed there.

We are not comfortable adding a case study from another region. We do not necessarily have the expertise or contextual knowledge required to provide a critical analysis of results from other areas. We prefer to thoroughly describe the specific context of our study, which we understand well, and allow readers to assess whether the proposed method is adaptable to their own context. Furthermore, including a few examples from other regions would not, in itself, demonstrate the method's universality across all contexts. However, if you believe this is absolutely necessary, we would be willing to make the effort.

19. Section 6.2: The EVT results for the observed data are currently presented late in the manuscript. I recommend moving this material to Section 2.2, as it fits naturally with the presentation of other PMP estimates (see my above comments about the EVT). For the manuscript flow, it is true that the EVT results should appear in the same table with the two other PMP estimation methods. It has now been moved to this section (Table 2 and L279-282).

20. L376–377 and for the second station later: Please summarize the GPD parameter estimates for each station in a table for clarity. The parameter estimates can now be found in Table 4. Thank you for the suggestion.

21. L416–417: The recommendations of the "National Academies of Sciences, Engineering, and Medicine" are specific to the Canadian or U.S. context. I suggest rephrasing such statements to make them more general and globally relevant. It is true that the recommendations address the North American context, but they stem from the broader observation that the definition and assumptions underlying PMP are outdated. This criticism is globally relevant, as the definition and assumptions of PMP are consistent worldwide. Therefore, we believe the recommendation can be considered general. Elements of this discussion have been added in the revised version of the manuscript (L400-403).

**EGUSPHERE-2024-2594**
**Detailed Responses to Reviewer's 2 Comments**

Anne Martin, Élyse Fournier and Jonathan Jalbert

May 29, 2025

I appreciate the authors' efforts in revising the manuscript, particularly their acknowledgment of the limitations of their proposed method for practical applications and their alignment with the National Academies of Sciences, Engineering, and Medicine (NASEM, 2024) recommendation to use an Extreme Value Theory (EVT)-based approach. However, the study's primary contribution lies more in its critical evaluation of PMP methodologies rather than in the statistical model itself. While previous studies have highlighted the flaws in the WMO-recommended PMP estimation methods, this study uniquely examines the limitations of physically based moisture maximization through a statistical framework before ultimately recommending an EVT-based approach. That said, I still have few concerns regarding the robustness of the conclusions, primarily due to the limitations of the EVT analysis and the inadequate discussion of sampling uncertainty.

Concerns:

1. The challenge of estimating very long return periods (e.g., 10,000 years) using only 75 years of data remains unresolved. Such extreme quantile estimates require substantial extrapolation, which increases uncertainty. Moreover, the stability of estimation may be sensitive to the threshold selection in the Peaks-Over-Threshold (POT) method. I suggest going over (NASEM, 2024) report about sampling uncertainty. The authors should explicitly discuss these uncertainties in the EVT approach to prevent misinterpretation by end-users. Otherwise, there is a risk of conveying an overconfident message about the reliability of these estimates.

   It is true that EVT-based PMP estimates are preferable, but they do not resolve all challenges. Extrapolating beyond the data range, especially for large return periods associated with PMP estimates, remains difficult and introduces substantial uncertainty. Such return level estimates should be accompanied by uncertainty evaluations (e.g., confidence intervals) to clearly communicate to end-users that PMP estimates carry substantial uncertainty inherent to extrapolation. The methodology based on simulated data, presented in Section 1.4 to address data scarcity, could also be adapted within the extreme value framework. Elements of this discussion have been added to the revised manuscript and confidence intervals have been added where it was not given in the previous version of the manuscript.

2. The authors acknowledge that identifiability issues affect the reliability of PMP estimates, yet the discussion remains largely theoretical. The manuscript would benefit from a practical demonstration in Section 6.3 of how alternative constraints—such as regularized maximum likelihood estimation or Bayesian priors—influence PMP estimates. If these methods were tested but found ineffective, the authors should clearly articulate why. This would strengthen the argument against using Pearson Type I for PMP estimation.

Thank you for the suggestion. We added two paragraphs in Section 6.3 (L368–381) describing the method we developed to address non-identifiability issues. While the method proved effective, we do not encourage its use because it introduces a high level of subjectivity into the analysis; something we aimed to avoid with the proposed statistical model for PMP.